# Programmable Sensor-System with Analog & Digital inputs (compatible with environmental sensors) and digital outputs.

Pranay Kamal Miriyala*, P Nitin Srinivas*, Tanish H Talapaneni*,
Thati Bhanoday*, and Bondugula Pranav*
*IIT Dharwad

## Abstract

This project centers on the creation of a versatile programmable sensor interface system. It accommodates various sensors, covering analog sensors' resistive, current, voltage, and capacitive measurements. Additionally, it supports I2C, SPI, and UART protocols for digital sensors. Sensors can be effortlessly integrated into their respective ports within the system. Data is stored locally in the microcontroller, allowing the Raspberry Pi to be configured for data collection, analysis, and internal storage. Moreover, the system offers the option of establishing a WiFi connection for remote data access.

This innovative system holds significant promise in the field of environmental monitoring. It covers many digital I2C environmental sensors, including the TSL25911FN Digital Ambient Light Sensor, MPU9250 Motion Sensor, LTR390-UV-1 UV Sensor, and SGP40 VOC Sensor. Additionally, various SPI-based environmental sensors like the BME280 measure temperature, humidity, and air pressure. Notably, the BME280 also supports I2C mode. The system's analog sensor ports are primarily dedicated to custom-designed sensors currently under research, in addition to available analog options such as the Capacitive Soil Moisture Sensor.

Beyond environmental sensing, this interface system has far-reaching applications in many industries and research fields. Biomedical research, automotive testing, geology, and seismology monitoring are prominent beneficiaries of this technological breakthrough. By seamlessly integrating sensors and providing robust data processing capabilities, this project substantially contributes to ongoing environmental monitoring and protection efforts. It offers a comprehensive solution for monitoring air, water, and soil pollution levels.

## 1  Introduction

We have engineered an advanced interface solution characterized by its adaptability in accommodating a diverse array of sensors. Our framework encompasses the fundamental infrastructure for interfacing with sensors, both analog and digital, thereby affording users the capability to integrate a broad spectrum of commercially accessible sensor technologies seamlessly.

In numerous application domains such as automation, medicine, health diagnostics, environmental assessment, and air quality monitoring, there exists a critical demand for the precise measurement of essential parameters. So, systems have been meticulously engineered to detect environmentally significant gases and track temperature and humidity levels using an amalgamation of environmental sensors (e.g., temperature, gas detection, moisture). To monitor individuals at risk, physiological sensors (e.g., heart rate, Blood pressure) and motion sensors become imperative. By incorporating enhanced flexibility in the selection of sensors, these systems are adept at furnishing such vital information to the end user or a designated individual tasked with monitoring the user.

Our proposed solution entails the creation of a system capable of receiving both analog and digital inputs and converting them into digital output. This system exhibits versatility in accommodating a diverse range of analog sensor categories, including resistive, capacitive, voltage, and current sensors. Additionally, digital sensors are compatible with UART, SPI, and I2C communication protocols. Furthermore, the system is designed with a programmable interface to afford users a high degree of customization and adaptability.

Our system simultaneously accommodates both analog and digital sensor inputs. The Resistive interface can handle sensors within the 5K$\Omega$ to 100 K$\Omega$ range, achieved with minimal hardware requirements of only two resistors and 2 PMOS components integrated into a single IC. Meanwhile, the Capacitor interface extends support to sensors ranging from 1 $\mu$F to 1000 $\mu$F, interfaced directly with the same microprocessor.

Voltage sensors with an output voltage within 5V can be readily integrated. Since most sensors in our survey operate below 5V, direct interfacing with the ADC pins is feasible. Furthermore, most commercially available current sensors are equipped with an I-V converter, allowing for seamless integration akin to voltage sensors. On the other hand, for digital sensors, our interface can handle up to 5 I2C sensors, 4 SPI sensors, and 2 UART sensors.

## 2 Goals

The programmable sensor caters to a variety of simple solutions for applications, including Power Monitoring, Materials Studies, Noise and vibration Testing, High Voltage Testing, Biomedical Research, Automotive Testing, Sonar, Radar and Ultrasonics, Geology/Seismology Monitoring. Our solution is to develop a system that takes in analog input and converts it into digital output. We aim to add flexibility to this system so that certain features like data modeling and prediction algorithms can be incorporated soon. We intend to make this digital output accessible in a user-friendly manner so that all can understand the applications of this system and can have a much broader scope.

Other Objectives:

- Separate channels for Resistive, Capacitive, and Digital-based sensors.

- Using WiFi-based transfer in Raspberry Pi for sending data to other devices connected to the WiFi network/cloud.

- GUI at the end user to provide flexibility to select the channels.

- Improving the range of resistive & capacitive sensors while using minimum hardware and consuming less power.

## 3 System Architecture and Design

| Component | Use |
|---|---|
| Raspberry Pi - 4B+ | Microprocessor used to store, process, and transfer data over WiFi. |
| Waveshare ADS1263 10- Ch 32-Bit ADC | Used for reading analog data from sensors |
| CD4052 4 Channel Multiplexer IC | Used for multiple digital SPI, I2C sensors |
| 5K$\Omega$, 16K$\Omega$ resistors | Used in resistive interface circuit |
| HCF4007UB DUAL COMPLEMENTARY PAIR | Used for PMOS - complementary switches for different resistive loads |
| $\mu$A741 General-Purpose Operational Amplifiers | Used for Op Amp based buffer: for connecting to ADC to reject noise from ADC to interface circuit |
| 1K$\Omega$, 10K$\Omega$ and 1.2M$\Omega$ resistors | Used in capacitive interface circuit |
| TL082 JFET-input Operational Amplifiers | Used for Op Amp based buffer: to prevent loading affect |
| I2C sensor ports(5) | To gather information from any I2C-based sensor. |
| SPI sensors ports(4) | To gather information from any SPI-based sensor. |
| UART sensors ports(2) | To gather information from any UART-based sensor. |

To incorporate digital sensors, we have used a couple of Multiplexers(one each for SPI and I2C) to connect multiple sensors simultaneously to collect different types of information in a parallel fashion. The UART sensors are connected directly to the UART ports.

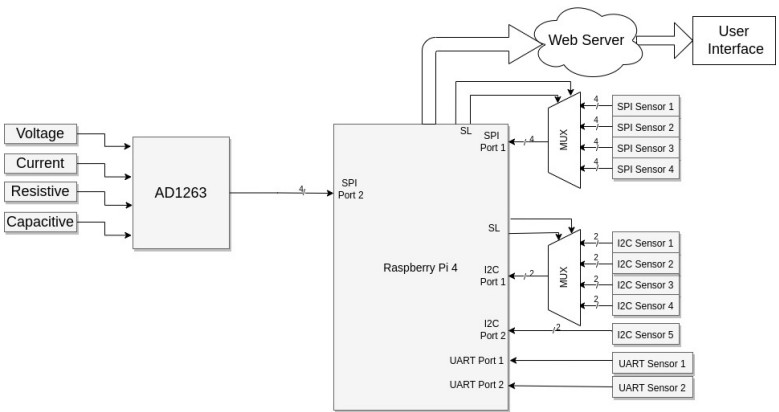

Figure 1: Overall system architecture: Analog and Digital

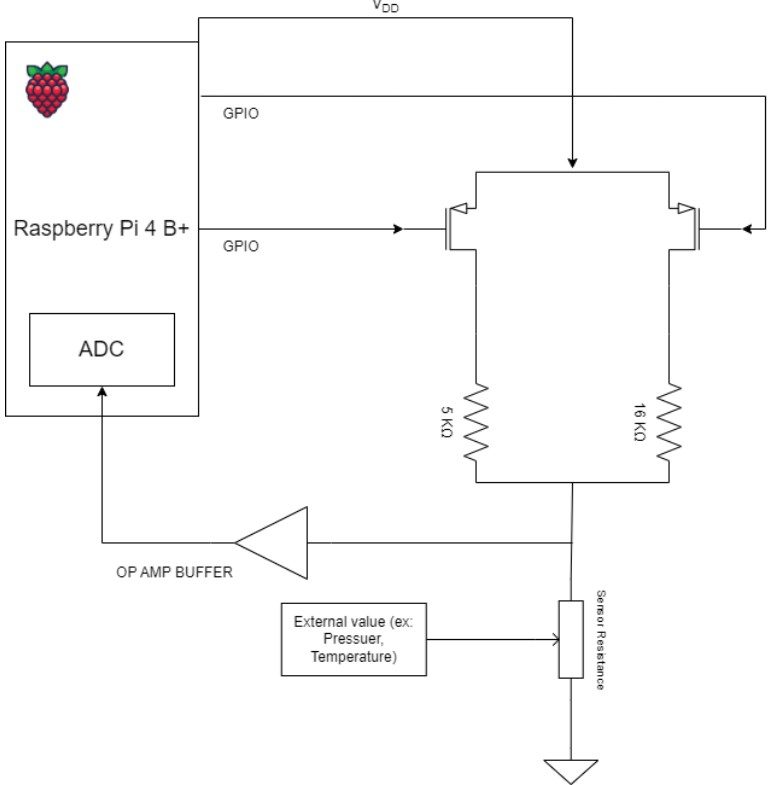

Figure 2: Resistive interface system: circuit level implementation

In Fig. 2, We have used a basic voltage divider circuit with a sensor as a resistive load, and complementary PMOS-based switches for changing the top resistance to extend the measurement range. GPIO pins of Raspberry Pi control the MOS switches, and the voltage divider equation, i.e. determine the resistance of the sensor,

$$V_{Out} = V_{DD} \cdot \frac{R_{Load}}{R_{Load} + R_{Top}} \tag{1}$$

The equation is simplified further into,

$$R_{Load} = \frac{R_{Top}}{\frac{V_{DD}}{V_{Out}} - 1} \tag{2}$$

The voltage sensed at the resistive divider is read through ADC, and since we know top resistance and output voltage, we can calculate load resistance.

The circuit can measure resistances in the range of 5KΩ to 20 KΩ for top resistance of 5 5KΩ. It can extend from 20 KΩ to 100 KΩ if the top resistance is switched to 16 KΩ.

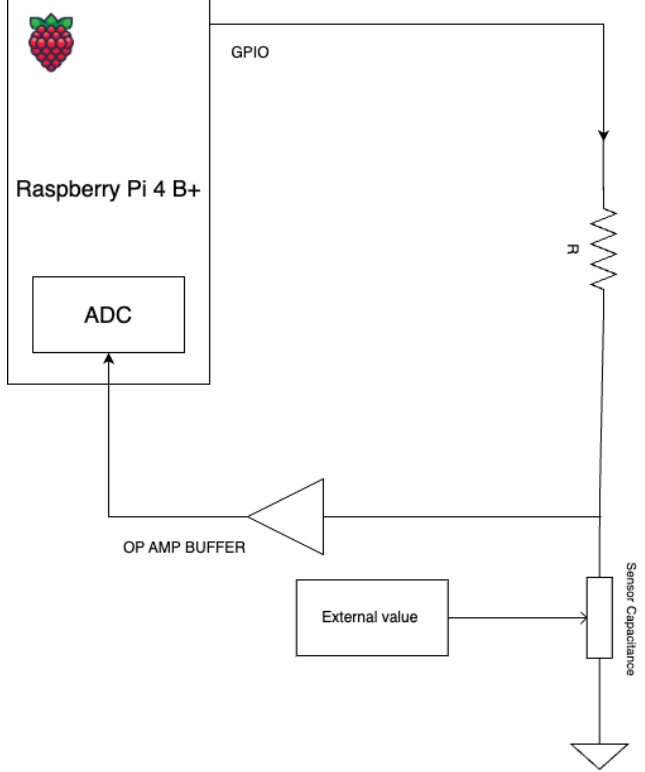

Figure 3: Capacitive interface system: circuit level implementation

In Fig. 3, we use a simple RC circuit to calculate the capacitance value. A buffer is placed at the junction between the resistor (R) and the capacitor (C) to isolate the specific node from the impedance of the subsequent circuit.

Initially, the GPIO pin of the Raspberry Pi is set to low for a certain amount of time, ensuring that the capacitor discharges completely. Then, the GPIO pin is set to high ( 3.3V ), causing the capacitor to charge exponentially.

The equation for the voltage across the capacitor is the above RC circuit is given by:

$$V(t) = V_0 \times \left(1 - e^{-\frac{t}{RC}}\right)$$

where $V_0 = 3.3$ V.

When the GPIO input pin is set high, the system captures the start time using the 'clock_gettime' function in the C programming language. Subsequently, when the voltage across the capacitor reaches 2 volts, the system records the end time.

Hence, from the above equation, we can find the value of C as all other values are known. As the time constant of the circuit cannot be arbitrarily low, three different values of resistors are used to cover the range of 1 $\mu$F to 1000 $\mu$F.

## 4   Addressing Challenges

The Key challenges were:

- Static Power consumption when using an interface with direct path from $V_{DD}$ to ground.

- Long range of values for Resistive circuits.

- Noise due to Raspberry Pi supply to the interfacing circuits.

- Noise dependent on sampling rate from ADC to the interfacing circuit.

- Loading affects the node present between the Resistor and Capacitor.

- Ensuring that the capacitive interface circuit works for a wide range

We have used PMOS switches for only the potential divider circuit when measuring the resistance value and switching it off when not in use. For a Longer range of values and retaining accuracy, we have used 2 top resistances with complementary PMOS switches, such that we get a larger range.

We have used decoupling capacitors for noise rejection from the Raspberry Pi supply. We have used an Op-Amp-based buffer circuit to isolate the ADC from the circuit.

To accommodate the loading effect, we isolated that node using a JFET-input op-amp-based buffer because the JFET-input op-amps have very high input impedance. To ensure that the capacitive interface circuit works for a wide range, we used three different resistances to adjust the time constant of the circuit.

## 5  Performance Evaluation and Testing Results

The resistive sensor interface circuit has been tested using LDR (Light Dependent Resistor).

Figure 4: Resistive results: Idle condition - ambient light - 10 K$\Omega$ + 2 K$\Omega$

Figure 5: Resistive results: Light condition

The capacitive sensor interface is tested for three different capacitor values, i.e., 1 $\mu$F, 100 $\mu$F, and 1000 $\mu$F. The resistance values taken are 1.2 M$\Omega$, 10 K$\Omega$, and 1 K$\Omega$, respectively, to ensure that the time constant is sufficiently high. The values of the capacitor obtained through the designed circuit and code are 1.049 $\mu$F, 109.21 $\mu$F, and 959.66 $\mu$F, all of which are within $\pm 5\%$ of the values obtained through LCR meter.

The Digital sensor interface was tested directly by running the Python code on the Raspberry Pi. The figure below gives the output of the sensor connected to channel 1. It also shows that there is no sensor connected to channel 4.

Figure 6: Digital sensor results

# 6 Concluding Remarks and Avenues for Future Work

Concluding Remarks:

- Our Programmable system ensures easy data accessibility via WiFi with the help of the inbuilt Raspberry Wi-Fi Chip.

- Multiple sensors can be connected at once with the help of 4:1 MUX).

- The data collected is high-resolution as the analog sensor data is converted to 32-bit digital data.

- We have covered three basic and standard communication protocols- UART, SPI, and I2C.

Avenues for Future Work:

- Extending the Core software to other kinds of microcontrollers, like RISC-V-based or ARM-based.

- Expanding the capacitance interface circuit's measurement range to include values as low as a few tens of pico farads while simultaneously enhancing measurement accuracy.

- Making a PCB hat by integrating resistive, capacitive, and digital sensor interfacing circuits for connecting with Raspberry Pi on ADC.

- We plan to keep a host of default sensors(covering a wide range of common parameters), and for configuring other custom sensors, the user will be allowed to add header files to the Raspberry Pi and configure the sensors

## 7   Availability

Source Code URL: GitHub Repository
Demo Video URL: Google Drive Link