# OpenReview forum: "Programmable Sensor-System with multi-channel sensor interface for both analog and digital inputs."
_helsinki.fi/ESPC/2023/Competition — ESPC 2023 ShortPresentation_

### Official Review · Reviewer_xRV8 · 2023-11-14

**Rating:** 3
**Confidence:** 3

**Summary:**

The authors develop and demonstrate a framework for interfacing with a variety of sensors. The system demonstrated has the capability to receive both analog and digital inputs and convert the received inputs into a digital output via three channels: one for resistive output, one for capacitive output, and one for digital output.

**Strengths:**

- The overall system architecture is provided in sufficient detail.
- The problem is timely and motivation is largely clear.
- The authors have provided some promising avenues for future work.
- The code is documented and well-structured, however the documentation can be improved.

**Weaknesses:**

- The presentation of the problem statement can be improved.
- There is limited discussion in the impact of variance in the voltages coming from the GPIO pins ($V_0$). For instance, how sensitive is the system when there are faults in the GPIO pins. The authors briefly mention the challenge in the noise due to PI supply, but the discussion is limited.
- Some analysis on the errors would have been useful. For instance, the authors mention specific ranges for capacitance and resistance but the evaluation results presented is for only a small sample in these ranges.
- Video can be improved by stating what you are doing, and why are you doing it in the method that is presented in the video; it can be made stronger by including the problem statement.  Specifically, details of what is the problem being addressed, what are the various components of the board, what the possible inputs and outputs, and what is the outcome of the experiment being conducted.

---

### Official Review · Reviewer_6UrX · 2023-11-18

**Rating:** 2
**Confidence:** 4

**Summary:**

This project aims to create a versatile programmable sensor interface system. The project aims to provide a solution to covering environmental sensors that output analog or digital signals. The solution is thought to cover analog sensors’ resistive, current, voltage, and capacitive measurements. The project claims it’s solution supports I2C, SPI, and UART protocols for digital sensors which can then be effortlessly integrated into their respective ports within the system (developed on a Raspberry Pi board).

**Strengths:**

The proposed system, i.e., solution entails the creation of a system that can receive both analog and digital inputs and convert them into digital output. In addition, the system introduces a programmable interface to afford users a high degree of customization and adaptability.

**Weaknesses:**

The report does not compare the proposed system with the existing works previously taken place by other people, thus, the novelty of the proposed solution remains unclear. Poor presentation of the work. The report misses the details about the electronic devices/utilities as well as the software.

---

### Official Review · Reviewer_EbaC · 2023-11-18

**Rating:** 2
**Confidence:** 3

**Summary:**

Authors have developed a versatile prototype for an IoT node using Raspberry Pi 4B that is programmable and can be interfaced with a wide variety of sensors.
This prototype has ports added for I2C sensors, SPI sensors, and UART sensors, along with the capabilities for distinguishing between resistive, capacitive, and digital sensors.

**Strengths:**

*The developed prototype has a unique capability of interfacing various sensors and thus, the it can be dynamically adapted to various usecases.

**Weaknesses:**

*Evaluation of the prototype is very weak. Authors should have evaluated certain usecases and shown how to use this prototype.
*One of the objectives mentioned is about GUI for the user. However, no such GUI details are presented in the report.

---

### Official Review · Reviewer_ph6q · 2023-11-20

**Rating:** 2
**Confidence:** 4

**Summary:**

This project focuses on demonstrating different interfaces to accommodate various sensors, covering analog sensors’ resistive, current, voltage, and capacitive measurements using Raspberry Pi.

**Strengths:**

Demos have been given to interface different sensors.

**Weaknesses:**

1. The demonstration could have been much better. It couldn't clearly motivate the project objective given the use of expensive and powerful Raspberry Pi.
2. Also, instead of one system, demos to interface different kind of sensors were done individually. This is very different than what is claimed in the document.
3. There is no novelty or innovation.

---

### Official Review · Reviewer_CCd1 · 2023-11-20

**Rating:** 2
**Confidence:** 3

**Summary:**

This project focused on the creation of a versatile programmable sensor interface system. To accommodates various sensors, covering analog sensors’ resistive, current, voltage, and capacitive measurements. It supported I2C, SPI, and UART protocols for digital sensors.

The authors goal was develop a system that takes in analog input and converts it into digital output. To add flexibility to this system so that  data modelling and prediction algorithms can be incorporated later. The project was well conducted with several team members working together. We appreciated the sharing of a video, the document, and making the software available in GitHub. There is a lack of a research question, which makes it difficult to assess the meaningfulness of the work. For the next work piece, we suggest the authors spend a little creating and documenting the experimental design.

**Strengths:**

The project was well conducted.
The paper showing a good System Architecture and Design.
They identified challenges due to noise and disturbances..
They conducted several experiments during the performance testing
A strength is making the source code available in GitHub Repository Demo Video URL: Google Drive Link
The video recording was in sync with the written explanation

**Weaknesses:**

The main weakness was the lack of a research problem and supporting literature review. It is difficult to ascertain the contribution of this work is to the general science of environmental sensing. Subsequently, there is no experimental design so the reviewer can follow the rationale behind the performance testing. This also affects the results and analysis section, which should reflect back on the research question.